# Neural Dynamics Self-Attention for Spiking Transformers

**Dehao Zhang[1], Fukai Guo[1], Shuai Wang[1], Jingya Wang[1], Jieyuan Zhang[1], Yimeng Shan[1],**
**Malu Zhang[1,2]\*, Yang Yang[1], Haizhou Li[2,3]**

[1]University of Electronic Science and Technology of China,

[2]Shenzhen Loop Area Institute, [3]The Chinese University of Hong Kong (Shenzhen)

## Abstract

Integrating Spiking Neural Networks (SNNs) with Transformer architectures offers a promising pathway to balance energy efficiency and performance, particularly for edge vision applications. However, existing Spiking Transformers face two critical challenges: (i) a substantial performance gap compared to their Artificial Neural Networks (ANNs) counterparts and (ii) high memory overhead during inference. Through theoretical analysis, we attribute both limitations to the Spiking Self-Attention (SSA) mechanism: the lack of locality bias and the need to store large attention matrices. Inspired by the localized receptive fields (LRF) and membrane-potential dynamics of biological visual neurons, we propose LRF-Dyn, which uses spiking neurons with localized receptive fields to compute attention while reducing memory requirements. Specifically, we introduce a LRF method into SSA to assign higher weights to neighboring regions, strengthening local modeling and improving performance. Building on this, we approximate the resulting attention computation via charge–fire–reset dynamics, eliminating explicit attention-matrix storage and reducing inference-time memory. Extensive experiments on visual tasks confirm that our method reduces memory overhead while delivering significant performance improvements. These results establish it as a key unit for achieving energy-efficient Spiking Transformers.

## 1 Introduction

Vision Transformers (ViTs) (Vaswani et al., 2017; Liu et al., 2021; Yu et al., 2022) achieve remarkable breakthroughs in computer vision tasks, including image classification (Chen et al., 2021; Deng et al., 2009), object detection (Carion et al., 2020; Shan et al., 2025), and semantic segmentation (Zhou et al., 2017; Xie et al., 2021; Yu et al., 2018). As the core of ViTs, the vanilla self-attention (VSA) computes query–key similarity through dot-product operation followed by softmax operation, incurring quadratic computational and memory costs with respect to the sequence length N (Yang et al., 2023). To address this issue, recent methods such as Linear attention (Katharopoulos et al., 2020; Zhang et al., 2024b) approximate or eliminate the softmax operation to explicitly compute the entire $N^2$ attention matrix, thus reducing memory usage and achieving linear-time complexity. However, these methods still rely on full-precision matrix multiplications, which incur substantial energy overhead and ultimately hinder their deployment on resource-constrained devices.

Spiking Neural Networks (SNNs) (Maass, 1997; Gerstner & Kistler, 2002; Masquelier et al., 2008) attract increasing attention due to their biological plausibility and potential for low-power computing. As the fundamental computational units, spiking neurons fire spikes only upon activation and remain silent otherwise, thereby enabling event-driven computation (Deng et al., 2020; Zhang et al., 2025d). This mechanism ensures sparse information transmission and effectively avoids redundant multiply-accumulate (MAC) operations, leading to reduced energy consumption and lower computational overhead (Caviglia et al., 2014; Roy et al., 2019). The combination of SNNs and Transformer architectures offers a potential pathway to balance energy efficiency and high performance (Zhou et al., 2023b; Huang et al., 2024; Yao et al., 2025; Wang et al., 2025a; Wang et al.).

---

\*Corresponding author: maluzhang@uestc.edu.cn

In recent years, several SNN-based Transformer architectures are proposed (Yao et al., 2024a; Shi et al., 2024; Cao et al., 2025; Wang et al., 2025b), including Spikformer (Zhou et al., 2023b), QK-Former (Zhou et al., 2024), and Spike-Driven-v3 (Yao et al., 2025). Nevertheless, SNN-based Transformers still exhibit a performance gap and incur substantial memory overhead. As shown in Fig. 1(a), the attention distributions of VSA and SSA differ markedly: SSA exhibits discrete and global patterns, whereas VSA shows localized and sparse ones. This mismatch hinders SNNs from focusing on specific regions of information. Moreover, as illustrated in Fig. 1(b), SNNs can leverage matrix aggregation and flexibly adapt their computational paradigms to different application scenarios. However, they still require explicit storage of large attention matrices, such as $\mathbf{QK}$ matrices of size $\mathrm{N}^2$ or $\mathbf{KV}$ attention matrices of size $\mathrm{d}^2$, which severely restricts deployment on resource-limited devices. Therefore, balancing the low-energy benefits of SNNs with the need for low memory overhead and high performance remains a critical open challenge.

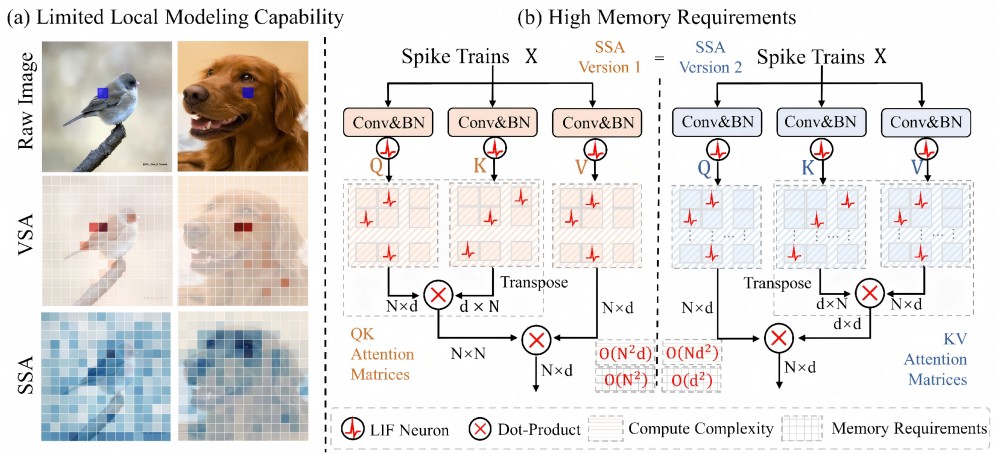

Figure 1: (a) Limited Local Modeling Capability: For a given n-th query (blue), VSA captures only limited and local relation. (b) High Memory Requirements: SSA requires explicit storage of their associated attention scores ($\mathbf{QK}$ or $\mathbf{KV}$), leading to substantial computational overhead.

Inspired by local receptive fields (Gaynes et al., 2022) and the temporal dynamics of neuronal membrane potentials, we propose LRF-Dyn, which enables attention computation via spiking neurons endowed with localized receptive fields. First, we theoretically and empirically compare VSA and SSA, showing that the lack of local modeling capacity in SSA accounts for its performance gap relative to VSA. To address this, we propose a Local Receptive Field (LRF) methods into SSA, assigning higher weights to spatial neighbors to strengthen locality. Building on this, we introduce LRF-Dyn, which establishes an approximate correspondence between self-attention aggregation and the charge–fire–reset dynamics of spiking neurons, thereby eliminating the need to explicitly store attention matrices. Extensive experiments across diverse vision tasks and architectures show that our method improves performance while further reducing inference-time memory requirements. The main contributions are as follows:

- We identify two major challenges in existing Spiking Transformers: (i) the performance gap caused by limited local modeling capability resulting from the removal of the softmax operation, and (ii) the high memory overhead due to the need to store attention scores.

- We propose LRF-Dyn to address these limitations. First, we incorporate a LRF method into SSA to assign higher weights to neighboring regions, thereby strengthening local modeling and improving performance. Building on this, we further approximate the computation via neuronal membrane-potential dynamics, eliminating the need to store attention matrices and reducing memory overhead during inference.

- Extensive experiments across diverse SNN architectures and visual tasks demonstrate that our method improves performance while further reducing inference-time memory requirements. It offers a practical solution for deployment in resource-constrained environments.

## 2 RELATED WORK

**Vision Transformer**: ViT (Dosovitskiy et al., 2020; Liu et al., 2022b) converts images into patch-based tokens and utilizes self-attention to establish global contextual relationships, facilitating the selective aggregation of informative features (Naseer et al., 2021; Yang et al., 2021). However, these models suffer from $\mathcal{O}(N^2)$ computational complexity and substantial memory overhead, which hinder scalability to large-scale visual tasks (Yang et al., 2023; Zhang et al., 2024a). To address these limitations, several studies (Choromanski et al., 2020; Guo et al., 2024) explore linear attention, which replaces the softmax operation with kernel-based approximations to reduce the complexity from $\mathcal{O}(N^2)$ to $\mathcal{O}(N)$. These methods significantly reduce both computational and memory requirements, making them more suitable for high-resolution images (Shen et al., 2021; Guo et al., 2022). However, existing models still rely on full-precision matrix multiplications, potentially increasing the model's energy consumption (Wang et al., 2020; Liu et al., 2022a).

**Spiking Transformer**: In recent years, researchers (Wang et al., 2023; Yao et al., 2023; Zhou et al., 2023a; Wei et al.) have explored combining SNNs with Transformers to achieve a trade-off between energy efficiency and performance (Yao et al., 2024b). Spikformer (Zhou et al., 2023b) introduces the SSA mechanism, which preserves spike-friendly properties while significantly improving the performance of SNNs. SpikingResformer (Shi et al., 2024) integrates ResNet with Transformer architectures to further reduce the model parameters. Furthermore, Spike-Driven-V3 (Yao et al., 2025) incorporates the Spike Frequency Approximation (SFA) mechanism into Spiking Transformers, enhancing their performance advantages. Although these models significantly reduce energy consumption, the inference process of SNN-based Transformers exhibits higher memory demands, limiting their deployment on resource-constrained devices (Aguirre et al., 2024; Zhang et al., 2025a).

## 3 PRELIMINARY

### 3.1 SPIKING NEURON MODEL

As the fundamental units of SNNs, spiking neurons (Izhikevich, 2003; Maass, 1997) receive presynaptic inputs and integrate them into the membrane potential, which is compared with the threshold to determine whether a spike is generated. Among them, the Leaky Integrate-and-Fire (LIF) neuron is the widely used model, whose dynamics are defined as follows:

$$\mathrm{U}[t+1] = \mathrm{H}[t] + \mathrm{W}S[t+1], \tag{1}$$

$$\mathrm{S}[t+1] = \Theta(\mathrm{U}[t+1] - \mathrm{V_{th}}), \tag{2}$$

$$\mathrm{H}[t+1] = \mathrm{V_{reset}}\mathrm{S}[t+1] + \tau\mathrm{U}[t+1](1 - \mathrm{S}[t+1]). \tag{3}$$

Here, $H[t]$ and $U[t]$ denote the pre-synaptic and post-synaptic membrane potentials, respectively, while $S[t]$ indicates the input spike at timestep $t$. $W$ denotes the synaptic weight matrix. Spike generation is defined by the Heaviside function $\Theta(\cdot)$: if a spike occurs ($S[t+1] = 1$), $H[t]$ is reset to $V_{reset}$; otherwise, $U[t+1]$ decays with time constant $\tau$ and updates $H[t+1]$. For clarity, we denote the above process as $\mathrm{SN}\{\cdot\}$, which represents the dynamics of spiking neurons.

### 3.2 SPIKING SELF-ATTENTION MECHANISMS

As the core component of the Spiking Transformer, the SSA mechanism captures spatio-temporal dependencies among tokens from spike trains and adaptively allocates importance across different regions. Given an input sequence $\mathbf{X} \in \mathbb{R}^{T \times B \times N \times D}$, it is projected through convolutional layers with distinct parameter matrices to obtain the Query $\mathbf{Q}$, Key $\mathbf{K}$, and Value $\mathbf{V}$ representations:

$$\mathbf{Q} = \mathrm{SN}\{\mathrm{BN}(\mathrm{Conv_Q}(\mathbf{X}))\}, \quad \mathbf{K} = \mathrm{SN}\{\mathrm{BN}(\mathrm{Conv_K}(\mathbf{X}))\}, \quad \mathbf{V} = \mathrm{SN}\{\mathrm{BN}(\mathrm{Conv_V}(\mathbf{X}))\}, \tag{4}$$

where $\mathrm{Conv}(\cdot)$ denotes a $1 \times 1$ convolution, and $\mathrm{BN}(\cdot)$ represents batch normalization. Inspired by the VSA mechanism, SSA computes the similarity via the dot product of $\mathbf{Q}$ and $\mathbf{K}$, and employs the resulting weights to aggregate $\mathbf{V}$. Specifically, the process of SSA is defined as follows:

$$\mathbf{Score} = \mathrm{s} \cdot \mathbf{Q} \times \mathbf{K}^{\top}, \quad \mathbf{Attn}' = \mathbf{Score} \times \mathbf{V}, \quad \mathbf{Attn} = \mathrm{SN}\{\mathbf{Attn}'\}, \tag{5}$$

s denotes the scaling factor. The attention output $\mathrm{Attn}'$ is processed by spiking neurons to ensure event-driven characteristics. Unlike VSA, SSA omits the softmax operation, thereby preserving the event-driven and spike-friendly characteristics of the attention mechanism.

## 4 PROBLEM ANALYSIS IN SPIKING TRANSFORMER

This section examines two main limitations of applying self-attention in SNNs: i) the omission of the softmax operation leads SSA to produce attention score distributions that deviate from those in VSA and ii) compared to other softmax-free attention variants, SSA introduces higher memory usage and inference overhead. Further details are provided in the following sections.

### 4.1 LIMITED LOCAL MODELING CAPABILITY

As the core mechanism of ViT, the VSA mechanism computes attention scores through the dot-product operation followed by the softmax operation. Specifically, given Query $\mathbf{Q} \in \mathbb{R}^{N \times C}$ and $\mathbf{K} \in \mathbb{R}^{N \times C}$, the attention score $\mathbf{Attn_{vit}}$ can be defined as follows:

$$\mathbf{Attn_{vit}} = \mathbf{softmax}\left(\frac{\mathbf{Q} \times \mathbf{K}^T}{\sqrt{d}}\right), \quad \mathrm{attn_{vit}}[i] = \frac{\exp\{q_i k_i\}}{\sum_{j=1}^{n} \exp\{q_i k_j\}}, \tag{6}$$

$d$ denotes the features dimension, and $\mathrm{attn_{vit}}[i] \in \mathbb{R}^{1 \times N}$ represents the attention score corresponding to between the $i$-th query $q_i$ and remaining tokens. When q and k are more similar, the attention score is higher. Since neighboring tokens usually exhibit stronger similarity, ViT demonstrates superior local modeling ability (Yang et al., 2021). As shown in Fig. 2, 76.8% of the attention scores in ViT are concentrated at short Manhattan distances. In contrast, SSA produces an almost uniform distribution of attention scores. This mismatch constrains the local modeling capacity of SSA and hinders its ability to capture spatial similarities among neighboring regions.

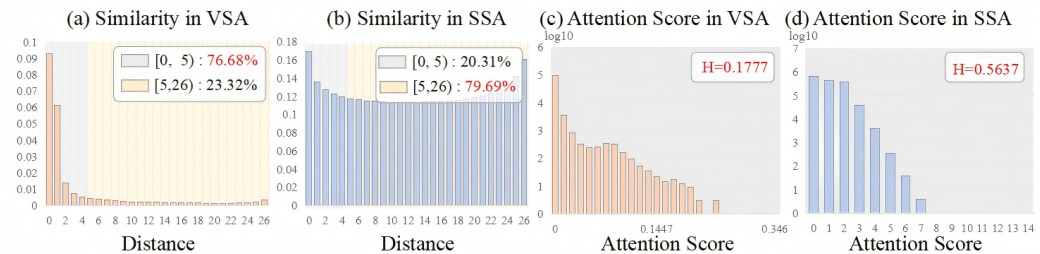

Figure 2: Mismatch between VSA and SSA attention scores: (a) and (b) show the average attention scores at different Manhattan distances, with VSA demonstrating stronger local modeling capabilities. (c) and (d) illustrate the distribution of attention scores, with VSA exhibiting lower entropy.

Furthermore, we compare the attention score distributions of SSA and VSA. As shown in Fig. 2, SSA exhibits a more uniform distribution, which hinders the model's ability to emphasize the relative importance of different regions. In contrast, VSA yields a lower-entropy distribution that focuses attention on a few critical regions, thereby enhancing feature extraction effectiveness.

### 4.2 HIGH MEMORY REQUIREMENTS DURING INFERENCE

Similar to other softmax-free models (Wang et al., 2020), SSA leverages the associative property of matrix operations to reduce computational complexity to $\mathcal{O}(Nd^2)$. Nevertheless, this computational benefit is accompanied by a pronounced increase in memory overhead during inference. Specifically, for the query $q_n[t]$ at timestep t, the attention score $\mathrm{attn_n}'[t]$ can be defined as follows:

$$\mathrm{attn}_n'[t] = \left(\sum_{j=1}^{N} q_n[t] k_j[t]^T e_j^T\right) \times v[t] = q_n[t] \times \sum_{j=1}^{N} k_j[t]^T v_j[t], \tag{7}$$

N denotes the total number of tokens, and $e_j$ is a column vector of length N with the $j$-th entry set to 1 and all others set to 0. As shown in Eq. 7, for the $n$-th token, it is necessary to store not only the $\mathbf{Q}$, $\mathbf{K}$, and $\mathbf{V}$ matrices at each timestep, but also the intermediate results of the $\mathbf{KV}$ multiplication, leading to an additional $\mathcal{O}(d^2)$ memory overhead. Notably, when $d = 512$, SSA incurs substantial memory demands. These limitations substantially hinder its deployment on resource-constrained devices, particularly on neuromorphic chips (Davies et al., 2018; Pei et al., 2019).

## 5 METHOD

In this section, we first incorporate the LRF mechanism into SSA to enhance local modeling and improve performance. Furthermore, we propose the LRF-Dyn that restructures this computations from the perspective of neuron modeling, reducing memory overhead while preserving performance.

### 5.1 SPIKING SELF-ATTENTION WITH LOCAL RECEPTIVE FIELDS

To address the limited local receptive field of spiking self-attention, we propose LRF-SSA, which integrates a local convolution module into the SSA mechanism to increase sensitivity to neighboring positions. Specifically, for the $n$-th token, its self-attention output $\mathrm{sattn}'_n[t]$ can be expressed as:

$$\mathrm{sattn}'_n[t] = \mathrm{q}_n[t] \times \underbrace{\sum_{j=1}^{N} k_j[t]^T v_j[t]}_{\text{Global Receptive Fields}} + \underbrace{\sum_d \sum_{i,j \in \Omega_d} r_{ij}^d \mathbf{V}^{\rho_k}}_{\text{Local Receptive Fields}}, \tag{8}$$

$\Omega_d = \{(i,j)|i,j \in \{-d,0,d\}\}$ represents the positional information of the neighboring region. To further reduce model complexity, we introduce multi-scale dilated convolutions to model local receptive fields. Specifically, two $3 \times 3$ depth-wise convolution kernels with dilation factors $d = 3$ and $d = 5$ are employed, where $r_{ij}$ denotes the convolutional parameter at position $(i,j)$.

**Theorem 1** *Let $i \in \mathcal{N} = \{1, \cdots, n\}$ denotes the token position and defined the Manhattan distance between two elements as $\Delta = d(i,j) = |i - j|$. The normalized attention weight of VSA is $\alpha_{ij}^{vsa} \propto \exp(-\beta\Delta)$. For SSA, the weight satisfies $\alpha_{ij}^{ssa} \propto (\alpha - \beta\Delta)_+$. The LRF-SSA is defined as $\alpha_{ij}^{lrf\text{-}ssa} = (1 - \lambda)\alpha_{ij}^{ssa} + \lambda r_{ij}$. Specifically, the expected receptive fields of LRF-SSA are defined:*

$$\mathbb{E}[\Delta_{\mathrm{lrf\text{-}ssa}}] = (1 - \lambda)\mu_{\mathrm{ssa}} + \lambda\mu_r, \quad \text{where} \quad \mu_{\mathrm{ssa}} = \mathbb{E}_{j \sim p_i^{\mathrm{ssa}}}[\Delta_{\mathrm{ssa}}], \quad \mu_r = \mathbb{E}_{j \sim p_i^r}(\Delta_r), \tag{9}$$

Since $\mu_{\mathrm{r}}$ represents the receptive field around each token, it naturally satisfies that $\mu_{\mathrm{r}} \leq \mu_{\mathrm{ssa}}$. Theorem 1 demonstrates that LRF-SSA preserves the local attention characteristics similar to those of VSA, whereas SSA exhibits uniformly distributed attention weights. This difference arises because SSA removes the softmax operation, resulting in a linear decay of attention scores and diminishing its ability to distinguish between neighboring positions. In contrast, LRF-SSA incorporates additional learnable weights within the spatial domain, allowing the model to concentrate more effectively on information captured by the local receptive field. We need to further evaluate whether this operation can effectively preserve the low-entropy distribution property of the softmax operation.

**Theorem 2** *For a given attention weights $x = (x_1, \ldots, x_N)$, the information entropy is defined as $H(x)$. In particular, the entropy of VSA is expressed as $H(\exp(-\beta\Delta))$, while SSA satisfies $H((\alpha - \beta\Delta)_+)$. LRF-SSA satisfies $H((1 - \lambda)p_{ij}^{ssa} + \lambda r_{ij})$. These entropies satisfy the ordering:*

$$\mathrm{H}(p_i^{\mathrm{lrf\text{-}ssa}}) \leq h(\alpha_i) + \alpha_i \mathrm{H}(p_i^{\mathrm{ssa}}) + (1 - \alpha_i)H(r_i) \leq \mathrm{H}(p_i^{\mathrm{ssa}}) \quad \text{where} \quad 0 \leq \alpha_i \leq 1, \tag{10}$$

Theorem 2 demonstrates that LRF-SSA exhibits a lower-entropy distribution more closely aligned with VSA. This effect arises primarily from the presence of local receptive field modules, which amplify the differences among attention scores. A detailed proof is provided in the Appendix C and Appendix D. Compared with VSA, our method eliminates the need for the softmax operation, thereby achieving performance comparable to SSA while reducing computational cost.

In summary, our method effectively preserves the local receptive field and low-entropy distribution characteristics of VSA. Compared with SSA, it demonstrates stronger local receptive field capability while introducing only minimal additional overhead (two 3×3 convolution kernels and an extra $\mathcal{O}(\mathrm{d}^2)$ computational cost). The effectiveness will be further validated in the experimental section.

### 5.2 IMPLEMENTING SELF-ATTENTION THROUGH NEURONAL DYNAMICS

To further reduce the inference-time memory footprint and computational latency of LRF-SSA, we propose LRF-Dyn. As previously noted, the LRF-SSA module must store the Q, K, and V matrices at each timestep, along with their corresponding attention scores. Specifically, when the model applies the SSA Version 2 illustrated in Fig. 1(b), LRF-SSA reduces the computational complexity to $\mathcal{O}(\mathrm{Nd}^2)$, while requiring the additional storage of an attention matrix of size $\mathrm{d}^2$.

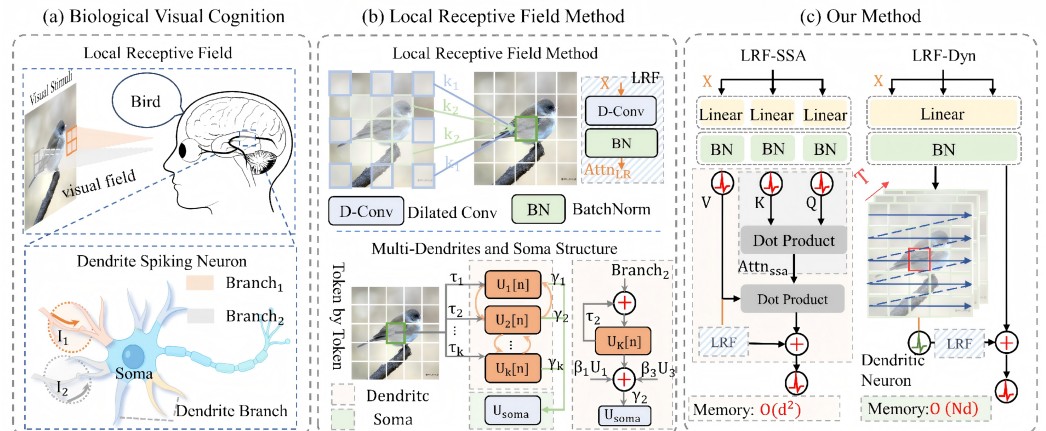

Figure 3: (a) Cognitive processes in biological vision, which exhibit local receptive field properties realized through multi-dendritic neurons. (b) The proposed LRF method together with the dynamic processes of dendritic neurons. (c) The implementation of LRF-SSA and LRF-Dyn.

Inspired by other softmax-free attention (Yang et al., 2023; Zhang et al., 2024b; Shen et al., 2021), LRF-SSA can be reformulated through causal inference to significantly reduce memory consumption. Specifically, Eq. 8 can be rewritten as follows:

$$\text{sattn}_n[t]' = q_n[t] \times \underbrace{\sum_{j=1}^{n-1} k_j[t]^T v_j[t] + k_n[t]^T v_n[t]}_{\text{Memory Potential}} + \underbrace{\sum_{d} \sum_{i,j \in \Omega_d} r_{ij}^d v_{\rho_k}[t]}_{\text{Presynaptic Input}}, \qquad (11)$$

In this manner, LRF-SSA method only needs to store $\sum_{j=1}^{n-1} k_j^\top v_j^\top$, thereby reducing the computational complexity $\mathcal{O}(Nd^2)$ to $\mathcal{O}(d^2)$. The attention output is then multiplied by the current query vector $q_n$ and transformed into a spike sequence through the SN layer. It closely parallels the charge–fire–reset dynamics of spiking neurons, where the first term represents membrane potential information and the second term represents presynaptic input.

Therefore, we propose LRF-Dyn which leverages neuronal dynamics to formulate a novel paradigm for self-attention computation, whose dynamical process can be defined as follows:

$$X_n[t] = \mathcal{A} \odot X_{n-1}[t] + \mathbf{\Gamma} \text{Token}_n[t], \quad \text{sattn}'_n[t] = X_n[t] + \sum_{d} \sum_{i,j \in \Omega_d} r_{ij}^d \cdot X_{\rho_k}[t], \qquad (12)$$

Here, $\text{Token}_n[t]$ denotes the token input at position $n$. $\mathcal{A} \in \mathbb{R}^d$ denotes the decay factor, and $\mathbf{\Gamma} \in \mathbb{R}^d$ is defined as the membrane capacitance constant. Inspired by the multi-timescale behavior of photoreceptor neurons (Zheng et al., 2024; Chen et al., 2024; Zhang et al., 2025b), we define $\mathcal{A}$ and $\mathbf{\Gamma}$ in a dendritic form with local receptive fields to better allocate attention scores across tokens:

$$\mathcal{A} = \underbrace{\begin{bmatrix} c_1, \\ c_2, \\ \vdots \\ c_{n-1}, \\ c_n \end{bmatrix}}_{\mathcal{C}}^T \times \begin{bmatrix} -\frac{1}{\tau_1} & \beta_{2,1} & 0 & \cdots & 0 \\ \beta_{1,2} & -\frac{1}{\tau_2} & \beta_{3,2} & \cdots & 0 \\ \vdots & \vdots & \ddots & \ddots & \vdots \\ 0 & 0 & \cdots & -\frac{1}{\tau_{n-1}} & \beta_{n,n-1} \\ 0 & 0 & \cdots & \beta_{n-1,n} & -\frac{1}{\tau_n} \end{bmatrix}, \quad \gamma = \begin{bmatrix} \gamma_1, \\ \gamma_2, \\ \vdots \\ \gamma_{n-1}, \\ \gamma_n \end{bmatrix}, \qquad (13)$$

Here, $d_n$ denotes the number of dendrites, and $\mathcal{C} \in \mathbb{R}$ represents the weights assigned to different dendrites. As shown in Fig. 3(b), for the $n$-th token, different dendritic branches produce distinct responses, which are further integrated by the soma through a specific mechanism to enhance spatial interactions and transform them into spike trains. Owing to the time-invariant property of the $\mathcal{A}$ matrix, the neuron can be trained efficiently following the (Chen et al., 2024). Compared with LRF-SSA, our model eliminates the SSA computation and only requires storing the membrane potential at each position, thereby substantially reducing memory usage during inference. In this study, n is set as 8. We will further validate the effectiveness of this method in the experimental section.

## 5.3 OVERALL ARCHITECTURE

In this section, we present the overall architectures of LRF-SSA and LRF-Dyn, respectively. Specifically, for an input spike train $x \in \mathbb{R}^{T \times N \times d}$, the computation of LRF-SSA is defined as follows:

$$\mathbf{Q}, \mathbf{K}, \mathbf{V} = \text{SN}\{\text{BN}(\text{Conv}(\mathbf{X}))\}, \quad \mathbf{Score} = \text{SN}\{s \cdot (\mathbf{Q} \times \mathbf{K}^T + \sum\nolimits_{i,j \in \Omega_d} r_{ij}^d) \times \mathbf{V}\}. \quad (14)$$

Compared with SSA, LRF-SSA introduces almost no additional parameters, yet it significantly enhances the local modeling. Building on this, LRF-Dyn further reduces the memory requirements during model inference. The dynamics of LRF-Dyn are defined as follows:

$$\mathbf{H} = \mathcal{F}^{-1}\{\mathcal{F}(\mathbf{K}) * \mathcal{F}(\mathbf{X})\}, \quad \mathbf{Score} = \text{SN}\{\sum \sum\nolimits_{i,j \in \Omega_d} r_{ij}^d \cdot \alpha_k \mathbf{H}_{p_k(t)}\}, \quad (15)$$

where $\mathcal{F}^{-1}$ denotes the forward and inverse Fourier transforms, respectively, and $*$ represents the convolution operation. $\mathcal{K}(t)$ is the convolution kernel, defined as $\Gamma\mathcal{C}\sum_{m=1}^{n-m}\mathcal{A}$. As shown in Fig. 3(c), both methods can be integrated into existing Transformer frameworks without any additional modifications. We further demonstrate the advantages of our approach in terms of both performance and memory efficiency in the experimental section.

## 6 EXPERIMENT

In this section, we compare with advanced SNN models on image classification (Deng et al., 2009) and semantic segmentation (Zhou et al., 2017) tasks. Additionally, we conduct ablation studies to validate the effectiveness of the proposed method, demonstrating its improved performance and reduced memory consumption. Detailed experimental results are provided below.

### 6.1 IMAGE CLASSIFICATION & SEMANTIC SEGMENTATION

**Image Classification Tasks**: we evaluate both LRF-SSA and LRF-Dyn on the ImageNet-1k dataset. Specifically, we substitute the SSA mechanism in existing Spiking Transformers, including Spikformer (Zhou et al., 2023b), QKFormer (Zhou et al., 2024), and Spike-Driven-V3 (Yao et al., 2025), with the proposed LRF-SSA and LRF-Dyn. Furthermore, we compare our approach with recently advanced SNN models (Fang et al., 2021; Hu et al., 2024; Yao et al., 2023; 2024b).

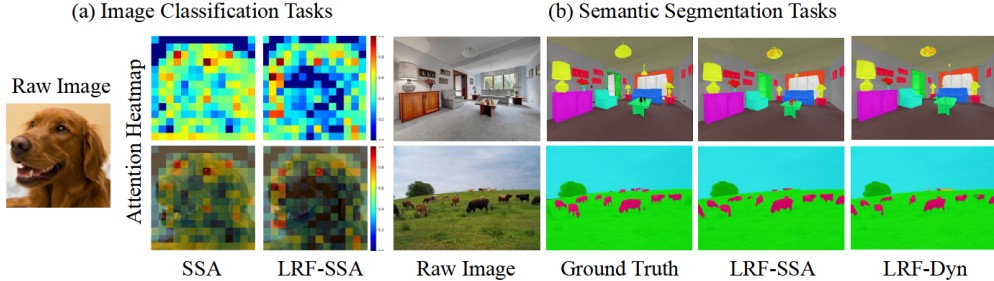

Figure 4: Visual results for image recognition and semantic segmentation. Both LRF-SSA and LRF-Dyn produce sparser attention scores and achieve finer-grained segmentation results.

As shown in Table 4, the proposed LRF-SSA method consistently delivers performance improvements across multiple architectures, with its two instantiations emphasizing different aspects. LRF-SSA primarily focuses on accuracy enhancement. On Spikformer, it improves accuracy by 1.24% and 0.85% at different parameter scales, while introducing fewer than 0.2M additional parameters. For QKFormer, LRF-SSA achieves an accuracy gain of 0.44% and 0.48% under the dim=384 and dim=512 configurations, respectively. Within the SDT-V3 architecture, LRF-SSA further demonstrates a favorable balance between efficiency and accuracy, attaining 76.22% recognition accuracy with only 5.24M parameters, substantially outperforming models of comparable size.

In contrast, LRF-Dyn preserves recognition performance while reducing storage complexity to $\mathcal{O}(\text{kd})$, where k denotes the number of dendrites. For instance, on SDT-V3, LRF-Dyn achieves

a gain of 0.82% and 0.44% while requiring significantly less inference memory. Moreover, compared with CNN-based SNN models (Fang et al., 2021; Hu et al., 2024), LRF-Dyn further delivers substantial performance gains without relying on attention mechanisms. As illustrated in Fig. 4(a), both LRF-SSA and LRF-Dyn exhibit ViT-like attention patterns but with sparser distributions, enabling the models to focus more effectively on salient regions. These results further demonstrate the superior performance and reduced memory requirements of our LRF-SSA framework.

Table 1: Comparison with Similar Methods on ImageNet-1K.

| Method | Architecture | SR. | Param.(M) | Acc.(%) |
|---|---|---|---|---|
| SEWResNet (Fang et al., 2021) | SEW-ResNet-34 | - | 21.79 | 67.04 |
| MSResNet (Hu et al., 2024) | MS-ResNet-34 | - | 21.80 | 69.42 |
| SDT-V1 (Yao et al., 2023) | Spike-driven-8-512 | $\mathcal{O}(Nd)$ | 29.68 | 74.57 |
| SDT-V2 (Yao et al., 2024b) | Meta-SpikeFormer-384 | $\mathcal{O}(d^2)$ | 15.08 | 74.10 |
| | Meta-SpikeFormer-512 | $\mathcal{O}(d^2)$ | 55.35 | 79.70 |
| Spikformer (Zhou et al., 2023b) | Spikformer-8-512 | $\mathcal{O}(d^2)$ | 29.68 | 73.38 |
| | Spikformer-8-768 | $\mathcal{O}(d^2)$ | 66.34 | 74.81 |
| **Spikformer + LRF-SSA** | Spikformer-8-512 | $\mathcal{O}(d^2)$ | 29.71 | **74.62 (↑1.24)** |
| | Spikformer-8-768 | $\mathcal{O}(d^2)$ | 66.53 | **75.66 (↑0.85)** |
| **Spikformer + LRF-Dyn** | Spikformer-8-512 | $\mathcal{O}(kd)$ | 29.71 | **74.51 (↑1.13)** |
| | Spikformer-8-768 | $\mathcal{O}(kd)$ | 66.53 | **75.58 (↑0.77)** |
| QKFormer (Zhou et al., 2024) | HST-10-384 | $\mathcal{O}(d^2)$ | 16.47 | 78.80 |
| | HST-10-512 | $\mathcal{O}(d^2)$ | 29.08 | 82.04 |
| **QKFormer + LRF-SSA** | HST-10-384 | $\mathcal{O}(d^2)$ | 16.55 | **79.24 (↑0.44)** |
| | HST-10-512 | $\mathcal{O}(d^2)$ | 29.18 | **82.52 (↑0.48)** |
| **QKFormer + LRF-Dyn** | HST-10-384 | $O(kd)$ | 16.44 | **79.21 (↑0.41)** |
| | HST-10-512 | $\mathcal{O}(kd)$ | 29.18 | **82.48 (↑0.44)** |
| SDT-V3 (Yao et al., 2025) | Efficient-Transformer-S | $\mathcal{O}(d^2)$ | 5.11 | 75.30 |
| | Efficient-Transformer-L | $\mathcal{O}(d^2)$ | 18.99 | 79.80 |
| **SDT-V3 + LRF-SSA** | Efficient-Transformer-S | $\mathcal{O}(d^2)$ | 5.24 | **76.22 (↑0.92)** |
| | Efficient-Transformer-L | $\mathcal{O}(d^2)$ | 19.25 | **80.31 (↑0.51)** |
| **SDT-V3 + LRF-Dyn** | Efficient-Transformer-S | $\mathcal{O}(kd)$ | 5.24 | **76.12 (↑0.82)** |
| | Efficient-Transformer-L | $\mathcal{O}(kd)$ | 19.25 | **80.24 (↑0.44)** |

* SR represents the storage complexity requirements during inference.

**Semantic Segmentation**: To evaluate the effectiveness of our methods, we further conducted experiments on more challenging segmentation tasks. In particular, we evaluate on the ADE20K dataset (Zhou et al., 2017), which consists of 20K training images and 2K validation images across 150 semantic categories. Following the experimental protocol of SDT-V3 (Yao et al., 2025), we evaluate our method on models with 5M and 19M parameters. As shown in Table 2, our approach achieves notable performance gains, improving by 2.6% and 2.2%, respectively. In addition, compared with ResNet (Yu et al., 2022) without attention modules, LRF-Dyn achieves higher performance (36.3% and 43.1%) while requiring fewer model parameters. As illustrated in Fig. 4, we further provide qualitative comparisons on selected examples. The proposed method produces more fine-grained segmentation results, whereas SSA tends to yield only localized segmentations. This observation provides additional evidence of the effectiveness of LRF-SSA.

Table 2: Performance of segmentation.

| Model | Para. (M) | Attn | T | MIoU(%) |
|---|---|---|---|---|
| ResNet | 15.5 | ✗ | 1 | 32.9 |
| (Yu et al., 2022) | 28.5 | ✗ | 1 | 36.7 |
| PVT | 28.2 | ✓ | 1 | 39.8 |
| (Wang et al., 2021) | 48.0 | ✓ | 1 | 41.6 |
| SDT-V3 | 5.1 + 1.4 | ✓ | 4 | 33.6 |
| (Yao et al., 2025) | 18.99 + 1.4[†] | ✓ | 4 | 41.3 |
| **SDT-V3** | 5.1 + 1.4 | ✓ | 4 | **36.2 (↑2.6)** |
| **+ LRF-SSA** | 10.0 + 1.4 | ✓ | 4 | **43.5 (↑2.2)** |
| **SDT-V3** | 5.24 + 1.4 | ✗ | 4 | **36.3 (↑2.7)** |
| **+ LRF-Dyn** | 19.25 + 1.4 | ✗ | 4 | **43.1 (↑1.8)** |

[†] Results reproduced by ourselves.

## 6.2 ENHANCED LOCAL MODELING ABILITY WITH LOWER MEMORY REQUIREMENTS

To further validate the effectiveness of our method, we follow (Zhang et al., 2025c; Ding et al., 2022) to visualize the receptive fields. As shown in Fig. 5(a), LRF-SSA significantly enhances the local receptive field of SSA and exhibits ViT-like local modeling capability. Similarly, LRF-Dyn also demonstrates a strong local modeling capacity. These results show the effectiveness of our method.

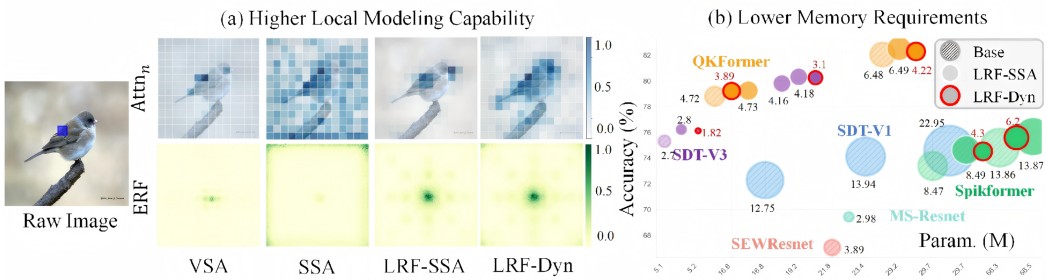

Figure 5: (a) Visualization of effective receptive field for different methods, where both LRF-SSA and LRF-Dyn demonstrate strong locality. (b) Comparative analysis of memory usage, accuracy, and parameter efficiency. The results show that LRF-Dyn maintains performance comparable to LRF-SSA while substantially reducing memory requirements during inference.

Furthermore, to compare the memory requirements of LRF-Dyn during inference, we visualize model accuracy and storage consumption at different scales. As illustrated in Fig. 5(b), Under the Spikformer-8-512 architecture, our method achieves a 1.13% increase in accuracy while simultaneously reducing memory usage by 49.4%. These results further confirm the effectiveness of our approach based on local receptive field enhancement and neuro-dynamics–inspired modeling.

## 6.3 ABLATION EXPERIMENT

To further substantiate the effectiveness of the proposed approach, we conduct comprehensive experiments on the CIFAR-100 dataset. Specifically, we investigate the respective contributions of local capability modeling and neurodynamics-inspired self-attention to both model performance and memory consumption, using the Spikformer architecture as the evaluation framework.

For the LRF module, we systematically examine the impact of varying the number of convolution kernels on the results. Notably, without the LRF module, LRF-SSA is equivalent to the SSA method. As shown in Table 3, increasing the kernel count consistently improves recognition accuracy, demonstrating that the introduction of local receptive fields enhances the performance of both LRF-SSA and LRF-Dyn. Furthermore, by comparing the LRF-Dyn approach with a causal SSA model, we observe that LRF-Dyn consistently demonstrates improved performance under the same conditions. This further supports the effectiveness of our method.

Table 3: Ablation Experiment.

| Method | w/o LRF | $\Omega \leq 1$ | $\Omega \leq 3$ | $\Omega \leq 5$ |
|---|---|---|---|---|
| LRF-SSA | 77.86 | 78.26 | 78.52 | **78.64** |
| LRF-Dyn | 77.78 | 78.16 | 78.50 | 78.57 |
| Caused SSA[†] | 74.30 | 75.30 | 76.20 | 76.50 |

[†] Results reproduced by ourselves.

## 7 CONCLUSION

In this paper, we analyze the challenges of integrating SSA into SNNs, focusing on two key limitations: the distinct attention distribution form compared to VSA, which limits further performance improvements, and the substantial memory overhead caused by the self-attention mechanism, which requires storing large attention weight matrices. To address these challenges, we propose LRF-Dyn, achieving an effective balance between performance and memory efficiency. First, we provide theoretical and empirical evidence for the importance of local modeling in self-attention, leading us to propose LRF-SSA, which incorporates a LRF module into SSA to enhance its local modeling capacity. Building on this, we approximate LRF-SSA with neuronal dynamics to remove the dependency on storing explicit attention matrices. This approach provides new theoretical insights and practical potential for deploying high-performance SNN models in resource-constrained edge environments.

## 8 ACKNOWLEDGMENTS

This work was supported by the National Natural Science Foundation of China (Grants 62576080 and 62220106008), the Sichuan Science and Technology Program (Grant 2024NSFTD0034), the Guangdong Introducing Innovative and Entrepreneurial Teams (Grant 2023ZT10×044), and the Shenzhen Science and Technology Research Fund (Grant JCYJ20220818103001002).

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

## A    USE OF LARGE LANGUAGE MODEL

In the preparation of this manuscript, we employ a Large Language Model (LLM) solely to aid and polish the writing. The LLM is used exclusively for language polishing, grammar correction, and improving the clarity and readability of the text. All technical ideas, methods, experiments, and conclusions presented in this paper are original contributions of the authors.

## B    TRAINING STABILITY OF LRF-DYN

To demonstrate the training stability of our LRF-Dyn, we plot the convergence curves of Spikformer and QKFormer on CIFAR-100. The results are as shown in Fig. 6.

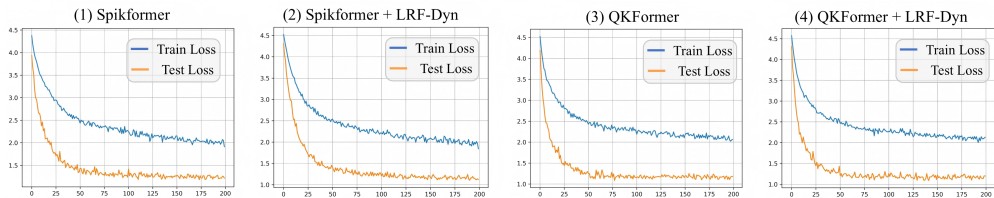

Figure 6: Comparison of convergence performance across different models.

It can be observed that our model exhibits a stable convergence pattern (Xu et al., 2023), which further demonstrates the effectiveness of our approach.

## C    PROOF OF THEOREM 1

In this section, we analyze the problem from the perspective of expected distance. Under an identical distance-bias assumption, SSA exhibits an expected distance that is almost global when compared with VSA. The LRF-SSA method partially alleviates the impact of this global receptive field.

### C.1    DEFINITION OF EXPECTED RECEPTIVE RADIUS

Given a sequence $x = \{x_1, x_2, \ldots, x_N\}$ of length $N$, we compute the expected receptive radius to measure the receptive field size under different attention paradigms, which is defined as follows:

$$\mu = \mathbb{E}[\Delta] = \sum_{j=1}^{N} \hat{\alpha}_{ij}\Delta, \qquad \Delta = d(i,j) = |i - j|, \tag{1}$$

where $\hat{\alpha}_{ij}$ denotes the normalized attention weight from position $i$ to position $j$, and $\Delta$ represents the distance between $i$ and $j$. Considering the spatial continuity of image sequences, we adopt the Manhattan distance as the metric. In this way, the expected receptive radius $\mu$ provides an effective measure of the receptive field across different attention paradigms. A larger value of $\mu$ indicates a larger effective receptive radius, corresponding to a more global receptive field.

### C.2    COMPARISON OF VSA, SSA AND LRF-SSA

Empirical studies in natural image statistics have demonstrated that patch similarity decreases as spatial distance increases (Zontak & Irani, 2011). For simplicity, we model this property as a linear decay with respect to distance. Specifically, for a given pair $(q_i, k_j)$, their similarity can be defined:

$$q_i^\top k_j \approx a - \beta\Delta, \tag{2}$$

Accordingly, for SSA, the expected receptive radius can be represented as follows:

$$\alpha_{ij}^{vsa} = \frac{q_i^\top k_j}{\sum_{j=1}^{N} q_i^\top k_j} \propto \exp\{-\beta|i - j|\}, \quad \mathbb{E}[\Delta] = \sum \Delta\alpha_{ij} = \Delta\frac{\exp\{-\beta\Delta\}}{\sum_{t=0}^{N}\exp\{-\beta t\}}, \tag{3}$$

As $n \to \infty$, the expected receptive radius of VSA can be defined as follows:

$$\mu_\infty^{vsa} = \frac{\exp(-\beta)}{1 - \exp(-\beta)} = \Theta(1). \tag{4}$$

Therefore, VSA exhibits a localized receptive field. In contrast, for SSA, since the softmax operation is omitted, its attention weight $\alpha_{ij}$ and the corresponding expected receptive radius can be defined:

$$\alpha_{ij}^{ssa} \propto (a - \beta\Delta)_+, \quad \mathbb{E}[\Delta] = \sum \Delta \alpha_{ij} = \frac{(\alpha - \beta\Delta)_+}{\sum_{t=0}^{N}(\alpha - \beta\Delta)_+} \tag{5}$$

As $n \to \infty$, the expected receptive radius of VSA can be defined as follows:

$$\mu_\infty^{ssa} = \frac{(N-1)(3\alpha - \beta(2N-1))}{3(2\alpha - \beta(N-1))} = \Theta(N). \tag{6}$$

Therefore, SSA exhibits a broader receptive radius than VSA.

As an SSA variant with local receptive fields, the attention weight of SSA is not only determined by $\alpha_{ij}$, but also incorporates an additional contribution from $r_{ij}$. Hence, the LRF-SSA attention weight can be expressed as a convex combination:

$$\alpha_{ij}^{\text{lra-ssa}} = (1 - \lambda)\alpha_{ij}^{ssa} + \lambda r_{ij}, \quad \mathbb{E}[\Delta_{\text{lra-ssa}}] = (1 - \lambda)\mu_{\text{ssa}} + \lambda\mu_r, \tag{7}$$

where $r_{ij} > 0$ denotes the local weights within the neighborhood $\Omega_d = \{j : |i - j| \leq d\}$, and $\mu_r \leq \mu_{\text{ssa}}$. Therefore, LRF-SSA exhibits an effective receptive field no larger than that of SSA, and this receptive field becomes increasingly local as $\lambda$ increases. In the extreme case where $\lambda > 1$, LRF-SSA tends toward a highly localized receptive field. Therefore, the receptive field radii of the three methods follow the relation:

$$\mu_{\text{VSA}} \leq \mu_{\text{LRF-SSA}} \leq \mu_{\text{SSA}} \tag{8}$$

Therefore, the proposed method exhibits local modeling capabilities more similar to SSA, while avoiding the use of the softmax operation.

## D PROOF OF THEOREM 2

In this appendix, we provide a detailed theoretical analysis of the entropy properties of different self-attention mechanisms. Benefiting from the softmax operation, VSA exhibits a sharp and low-entropy distribution, concentrating most of the attention mass on a few neighboring positions. In contrast, SSA produces a more dispersed distribution, leading to higher entropy.

### D.1 INFORMATION ENTROPY

From an information-theoretic perspective, the entropy of the attention distribution provides a natural measure of its uncertainty and sharpness. Specifically, for a given attention weight vector $x = (x_1, \ldots, x_n)$, the entropy is defined as:

$$H(x) = -\sum_{i=1}^{n} x_i \log x_i, \tag{9}$$

where we use natural logarithms. A larger entropy indicates a more uniform and smoother distribution of attention scores, whereas a smaller entropy corresponds to a more concentrated and sharper allocation of attention.

### D.2 ENTROPY OF VSA, SSA AND LRF-SSA

As shown in Eq. 17, the similarity decays linearly with distance $\Delta$. Therefore, for VSA, the induced truncated geometric distribution over distances (ignoring multiplicity) is:

$$P_{\text{vsa}}(\Delta) = \frac{\exp\{-\beta\Delta\}}{\sum_{t=0}^{N-1}\exp\{-\beta t\}} = \frac{(1-r)r^\Delta}{1-r^N}, \qquad r = \exp\{-\beta\} \in (0, 1), \tag{10}$$

discretized over $\Delta = 0, \ldots, N-1$. The corresponding entropy admits the closed form:

$$H_{\text{vsa}}(N, r) = \log \frac{1 - r^N}{1 - r} - \frac{r}{1 - r} \cdot \frac{1 - Nr^{N-1} + (N-1)r^N}{1 - r^N} \ \log r, \tag{11}$$

and in the infinite-length limit, we obtain:

$$H_{\text{vsa}}(\infty, r) = -\log(1 - r) - \frac{r}{1 - r} \log r = \mathcal{O}(1). \tag{12}$$

Thus VSA yields a bounded, low-entropy distribution independent of $N$.

In contrast to the truncated geometric distribution of VSA, SSA employs linearly decaying weights without softmax normalization. Specifically, the unnormalized score is modeled as:

$$w(\Delta) = \alpha - \beta\Delta, \qquad 0 \le \beta \le \frac{\alpha}{N-1}, \tag{13}$$

where the constraint ensures non-negativity across all positions. After normalization, the distance distribution becomes:

$$P_{\text{ssa}}(\Delta) = \frac{\alpha - \beta\Delta}{\alpha N - \beta \frac{N(N-1)}{2}}, \qquad \Delta = 0, \ldots, N-1, \tag{14}$$

which leads to the entropy:

$$H_{\text{ssa}}(N, \alpha, \beta) = -\sum_{\Delta=0}^{N-1} \frac{\alpha - \beta\Delta}{S_0} \left[\log(\alpha - \beta\Delta) - \log S_0\right], \qquad S_0 = \alpha N - \beta \frac{N(N-1)}{2}. \tag{15}$$

For the special case $\beta = 0$, $P_{\text{ssa}}$ reduces to the uniform distribution and the entropy attains its maximum $H_{\text{ssa}} = \log N$. At the other extreme, when $\beta = \alpha/(N-1)$, the distribution becomes triangular and the entropy satisfies $H_{\text{ssa}} = \log N + \mathcal{O}(1)$. Therefore, the entropy of SSA scales as:

$$H_{\text{ssa}} = \Theta(\log N), \tag{16}$$

indicating a high-entropy distribution that spreads broadly across the sequence and corresponds to a nearly global receptive field.

Finally, we analyze the entropy of LRF-SSA. LRF-SSA can be formulated as a convex combination of SSA and a more concentrated local distribution $R_i$:

$$P_i^{\text{lra-ssa}}(\Delta) = (1 - \lambda_i) P_i^{\text{ssa}}(\Delta) + \lambda_i R_i(\Delta), \qquad \lambda_i \in [0, 1]. \tag{17}$$

The entropy of LRF-SSA is thus:

$$H\left(P_i^{\text{lra-ssa}}\right) = -\sum_{\Delta=0}^{N-1} \left[(1 - \lambda_i) P_i^{\text{ssa}}(\Delta) + \lambda_i R_i(\Delta)\right] \log \left[(1 - \lambda_i) P_i^{\text{ssa}}(\Delta) + \lambda_i R_i(\Delta)\right]. \tag{18}$$

Although a closed-form solution is difficult to obtain, we can derive an information-theoretic upper bound by introducing an auxiliary Bernoulli variable $Z \sim \text{Bernoulli}(\lambda_i)$ and applying the chain rule of entropy:

$$H\left(P_i^{\text{lra-ssa}}\right) \le h(\lambda_i) + (1 - \lambda_i) H(P_i^{\text{ssa}}) + \lambda_i H(R_i), \tag{19}$$

where $h(\cdot)$ denotes the binary entropy. Since $R_i$ is designed to be more localized, we typically have $H(R_i) \le H(P_i^{\text{ssa}})$, which implies:

$$H\left(P_i^{\text{lra-ssa}}\right) \le H\left(P_i^{\text{ssa}}\right). \tag{20}$$

Therefore, LRF-SSA consistently produces lower entropy than SSA, and the degree of reduction is controlled by the mixing coefficient $\lambda_i$; in other words, LRF-SSA interpolates between the global high-entropy behavior of SSA and the localized low-entropy property of VSA, providing a flexible trade-off between global coverage and local sharpness.

