# OpenReview forum: "Neural Dynamics Self-Attention for Spiking Transformers"
_ICLR.cc/2026/Conference — ICLR 2026 Poster_

### Official Review · Reviewer_MjDS · 2025-10-16

**Soundness:** 3
**Presentation:** 3
**Contribution:** 3
**Rating:** 6
**Confidence:** 4

**Summary:**

### 1. Research Problem

*  Existing Spiking Transformer models underperform compared to their ANN counterparts, primarily due to limitations in the spiking self-attention (SSA) mechanism which lacks effective attention localization.

*  SSA requires explicit storage of large attention matrices (such as QK and KV), resulting in significant memory overhead, especially during inference on resource-constrained hardware.

*  Without the softmax operation, SSA generates nearly uniform attention distributions, losing the ability to focus on spatially relevant regions, which is critical in visual tasks.


### 2. Proposed Method

* LRF-SSA (Local Receptive Field Self-Attention):

  * Enhances SSA by incorporating multi-scale dilated convolutions that introduce spatial bias through local receptive fields.
  * Adjusts attention weights to favor nearby tokens, improving local modeling capability while preserving spike-driven characteristics.
  * The attention weights are computed as a convex combination of SSA attention and a local convolutional kernel:
    α_lrf-ssa = (1 - λ) * α_ssa + λ * r_ij

* LRF-Dyn (Local Receptive Field with Neural Dynamics):

  * Reformulates attention using principles of spiking neuron dynamics, simulating membrane potential updates and dendritic processing.
  * Avoids explicit computation and storage of QK or KV matrices by using accumulated membrane states, reducing memory complexity.
  * The attention computation becomes a charge–fire–reset-like process that aligns with the biological plausibility of SNNs.


### 3. Theoretical Contributions

* Entropy-based attention analysis:

  * Demonstrates that LRF-SSA produces lower-entropy attention distributions than SSA, which better reflect the selective nature of softmax-based VSA.

* Expected receptive field radius: Introduces a formal measure of attention locality and proves that LRF-SSA reduces the average receptive field size compared to SSA, while remaining closer to the behavior of VSA.

* Neural dynamics formulation of attention: Establishes an equivalence between spiking attention and dynamic state updates in multi-dendritic neurons, paving the way for biologically grounded attention mechanisms.

* Memory and computation complexity: Reduces inference-time memory from O(d²) to O(kd), with k representing the number of dendritic branches, offering theoretical scalability for edge deployment.


### 4. Experimental Results

* Image classification (ImageNet-1k): LRF-SSA improves top-1 accuracy by up to 1.24% with negligible parameter increase (<0.2M). LRF-Dyn achieves similar or slightly reduced accuracy (within 0.1%) while cutting memory usage to O(kd).

* Semantic segmentation (ADE20K): LRF-SSA improves mIoU by 2.6% and LRF-Dyn by 1.8% over the SSA baseline, showing stronger spatial discrimination with fewer parameters than attention-free baselines like ResNet.

* Ablation studies (CIFAR-100): Performance scales with number of convolution kernels (larger Ω yields higher accuracy), validating the contribution of local spatial modeling.

**Strengths:**

* The paper introduces a novel perspective on spiking self-attention by recasting attention as a neural dynamic process, drawing from the charge–fire–reset behavior of biological neurons. This is a conceptually original contribution that bridges the gap between algorithmic attention mechanisms and neurophysiological processes.

  * The integration of localized receptive fields into spike-based self-attention (LRF-SSA) is a creative fusion of biologically inspired modeling and modern deep learning practice. It departs from the conventional global attention view in SNNs and enables spatial bias without softmax.

  * The work removes a critical limitation in existing spiking Transformers by eliminating the need for explicit QK/KV matrix storage, allowing for scalable inference on neuromorphic or memory-limited hardware—an underexplored but increasingly relevant design constraint.

  * The experimental content is detailed and clear, and the effectiveness of the method is verified on different tasks.

**Weaknesses:**

* Limited theoretical depth for LRF-Dyn: While the paper presents a compelling neuro-dynamic approximation of self-attention, the theoretical treatment of LRF-Dyn remains less rigorous than that of LRF-SSA. Specifically, the membrane potential update formulation lacks an explicit connection to the global attention weighting mechanism it replaces.


* Experiments limited to static vision benchmarks: Despite the motivation from real-time and neuromorphic computing, the evaluation is restricted to frame-based, static datasets (ImageNet-1k and ADE20K). These benchmarks do not fully capture the event-driven potential or energy-efficiency implications of spiking systems. The paper would benefit from experiments on neuromorphic datasets (e.g., N-Caltech101, DVS Gesture, or N-Cars) to substantiate claims of biological plausibility and resource efficiency.

* Experimental performance improvement is limited.

**Questions:**

* Can the authors provide a more rigorous theoretical analysis of LRF-Dyn? Specifically, is there a formal approximation bound or information-preserving guarantee that connects LRF-Dyn to standard attention mechanisms?

* How does the proposed method handle long-range temporal dependencies inherent to spiking inputs? Would LRF-Dyn remain effective in tasks requiring fine-grained temporal precision, such as DVS-based continuous input streams?

* Have the authors considered evaluating the method on neuromorphic datasets?

*  Does the LRF-Dyn formulation introduce stability issues during training, such as exploding or vanishing membrane potentials? Can the authors provide convergence plots or empirical analysis to support its robustness?

* Please add some missing citations, eg:  https://arxiv.org/abs/2311.09376

---

> ### Author Response · Authors · 2025-11-21
>
> We sincerely appreciate your recognition of our work. In response to the questions, we provide the following clarifications. All suggested revisions will be incorporated into the revised version to further enhance its clarity and presentation
> # Weakness 1 & Question 1:  Limited theoretical depth for LRF-Dyn
> Thank you for your positive feedback on our work. First, we provide a detailed explanation of the membrane potential update mechanism in LRF-Dyn. Second, we clarify its correspondence to LRF-SSA.
>
> ## Dynamics of Membrane Potential Updates
> Given an input $\text{Token} \in \mathbb{R}^{T \times B \times N \times D}$, LRF-Dyn converts it into spike trains through its multi-dendrite and soma structures. The multi-dendrite component extracts features from each input $\text{Token}[t]$, while the soma integrates the responses from all dendritic neurons and transforms them into spike trains.
> For clarity of notation, we restrict our discussion to the input at the $n$-th position, denoted by $\text{Token}_n[t]$. Dynamics of the multi-dendrite structure can then be expressed as follows:
>
> $$I_{dendritc_n}[t] = \mathcal{A} \odot I_{dendritc_{n-1}}[t] + \Gamma Token_n[t], \quad I_{lrf_n}[t] = I_{dendritc_n}[t] + \sum_d \sum_{i, j} r_{ij}^d I_{dendritc_{\rho_k}}[t], $$
>
> $I_{\text{dendritic}\_n}[t]$ denotes the postsynaptic membrane potential of the dendrite at the n-th position, and $I_{\text{lrf}_n}[t]$ represents the presynaptic membrane potential of the soma equipped with a local receptive field. $\rho_k$ specifies the $\Omega \le 3$ neighborhood around the $n$-th position. Finally, the outputs of all dendritic branches are integrated by the soma structure:
>
> $$U_n[t] = H_n[t-1] + I_{lrf_n}[t], $$
>
> $$S_n[t] = \Theta (U_n[t] - V_{th}), $$
>
> $$H_n[t]= V_{reset} S_n[t] + \tau U_{n-1} (1 - S_n[t] ),$$
>
> $V_{\text{reset}}$ denotes the reset membrane potential, and $V_{\text{th}}$ denotes the firing threshold. In this manner, the membrane potential dynamics can be converted into a spike-based representation.
> ## Relationship between LRF-Dyn and LRF-SSA
> As illustrated in Eq.11 and Eq.12, the LRF-SSA process for the $n$-th token can be described as follows:
>
> $$sattn'\_n[t] = \text{q}\_n[t] \times \sum_{j=1}^N \text{k}\_j[t]^T \text{v}\_j[t] + \sum_d \sum_{i,j\Omega_d}r\_{ij}^d \textbf{V}\_{\rho\_{k}},$$
>
> Subsequently, $\text{sattn}'\_n[t]$ is input into an LIF neuron to produce a spike-based representation. The dynamics of the LIF neuron are defined as follows:
>
> $$U\_n[t] = H\_n[t-1] + sattn'\_n[t], $$
> $$S\_n[t] = \Theta (U_n[t] - V_{th}), $$
> $$H_n[t]= V_{reset} S_n[t] + \tau U_{n-1} (1 - S_n[t] ),$$
>
> Here, $\mathbf{q}\_n[t]$, $\mathbf{k}\_n[t]$, and $\mathbf{v}\_n[t]$ denote the query, key, and value at the $n$-th position, respectively. $\sum_{j=1}^N \mathbf{k}\_j[t]^{\top} \mathbf{v}\_j[t]$ is approximated by the membrane potential in LRF-Dyn, while $\mathbf{q}_n[t]$ corresponds to the multi-dendrite weighting $\mathcal{A}$ in LRF-Dyn. In this way, we establish an approximate correspondence between LRF-Dyn and LRF-SSA.
>
> # Weakness 2: Experiments on Neuromorphic Benchmarks
> Following [1], we conduct evaluations on N-Caltech101, DVS Gesture, and NCARS. The results are summarized as follows:
> |Method|T|N-Caltech101|DVS Gesture|NCARS|
> |:-:|:-:|:-:|:-:|:-:|
> |Spikformer [1]|10|78.9|96.5|93.2|
> |+LR-Attn|10|81.2|97.3|95.7|
> |+LR-SSA|10|81.1|97.4|95.7|
>
> [1] Spikformer: When spiking neural network meets transformer. ICLR 2023.
>
> # Weakness 3: Experimental performance improvement is limited.
> This is primarily because QKFormer already demonstrates strong baseline performance on image classification tasks, leaving limited room for further improvement. To further evaluate its effectiveness, we extend our study to the more challenging image segmentation task, which requires stronger visual long-sequence modeling [2]. The experimental setup follows the Spike-Driven-v3 configuration [3], and the results are presented below:
>
> |Method|Para.|Acc|
> |:-:|:-:|:-:|
> |QKFormer|17.83|34.65|
> |QKFormer+LRF-SSA|17.85|37.19|
> |QKFormer+LRF-SSA|17.85|37.16|
>
> The experimental results show that both LRF-SSA and LRF-Dyn achieve substantial performance improvements of 2.54% and 2.51%, respectively, further validating the effectiveness of the proposed method.
>
> [2] Vision Transformers for Dense Prediction, ICCV 2021.
>
> [3] Scaling Spike-Driven Transformer With Efficient Spike Firing Approximation Training, T-PAMI 2025.

---

> > ### Author Response · Authors · 2025-11-21
> >
> > # Question 2: Limitations in Capturing Long-Range Temporal Dependencies
> > As you noted, long-term dependency is indeed a central challenge for spiking neurons. As the core of LRF-Dyn, multi-dendritic neurons have been shown in prior studies [4, 5] to effectively address this issue. They capture multi-scale information through parallel dendritic branches, while the soma integrates signals from different dendrites. The somatic membrane potential consequently preserves long-range temporal information. As shown in the following, multi-dendritic neurons exhibit superior performance on long-sequence tasks.
> >
> > |Method|S/PS-MNIST|SHD|
> > |:-:|:-:|:-:|
> > |LIF|74.91/89.28|71.40|
> > |TC-LIF [4]|96.46/97.35|91.35|
> > |DH-LIF [5] |98.9/94.52|91.34|
> >
> > [4] TC-LIF: A Two-Compartment Spiking Neuron Model for Long-Term Sequential Modelling, AAAI 2025.
> >
> > [5] Temporal dendritic heterogeneity incorporated with spiking neural networks for learning multi-timescale dynamics. Nature Communications 2024.
> >
> > # Question 3: Evaluation on Neuromorphic Datasets
> > To address your concern, we conducted additional experiments on DVS-based benchmarks, where the method demonstrated improved recognition performance and more accurate temporal discrimination.
> >
> > |Method|T|N-Caltech101|DVS Gesture|NCARS|
> > |:-:|:-:|:-:|:-:|:-:|
> > |Spikformer [3]|10|78.9|96.5|93.2|
> > |+LR-Attn|10|81.2|97.3|95.7|
> > |+LR-SSA|10|81.1|97.4|95.7|
> >
> > # Question 4: Stability and Convergence of LRF-Dyn
> > To further address your concern, we provide the training curves for the entire optimization process. As shown in Fig.6 in the appendix, our method exhibits stable convergence across multiple architectures.
> >
> > # Question 5: Please add some missing citations, eg: https://arxiv.org/abs/2311.09376
> > We sincerely appreciate your suggestion, and we will incorporate additional relevant references in the revised manuscript.

---

### Official Review · Reviewer_Ttxv · 2025-10-26

**Soundness:** 1
**Presentation:** 2
**Contribution:** 2
**Rating:** 2
**Confidence:** 5

**Summary:**

This paper addresses the issue of unfocused global attention and high memory overhead in spiking transformers. To resolve these problems, this paper proposes Local Receptive Field Spiking Self-Attention (LRF-SSA), which strengthens local attention by assigning higher weights to spatially adjacent positions. Building on LRF-SSA, the paper also introduces LRF-Dyn, a method that eliminates the need for explicit attention-matrix storage and reduces memory overhead.

**Strengths:**

1. Theoretical analyses are provided to support the proposed LRF-SSA mechanism.
2. The proposed LRF-Dyn maintains performance comparable to LRF-SSA while reducing memory requirements during inference.

**Weaknesses:**

1. The motivation behind LRF-SSA requires further validation. The paper points out a mismatch in attention patterns between SSA and VSA: as shown in Fig. 1(a) and Fig. 2, VSA emphasizes local relations, while SSA exhibits unfocused global attention. However, this lack of local attention in SSA is a result of its inherent limitations rather than the root cause of its shortcomings. Therefore, enhancing local attention may not fundamentally resolve the issue. In fact, this issue stems from information loss due to the binary nature of spikes in SSA. Previous research [1] has identified this problem and proposed solutions that effectively focus on key features. As a result, the motivation for this paper needs reconsideration and further validation.
2. The proposed method achieves only marginal accuracy improvements on the ImageNet-1k classification task, particularly on the state-of-the-art architecture QKFormer, where the accuracy gain is less than 0.5%.

[1] Xiao, Y., et al. (2025). Rethinking Spiking Self-Attention Mechanism: Implementing a-XNOR Similarity Calculation in Spiking Transformers. Proceedings of the Computer Vision and Pattern Recognition Conference.

**Questions:**

Please compare existing methods and explain why LRF is more effective than these approaches in addressing the limitations of SSA.

---

> ### Author Response · Authors · 2025-11-21
>
> # Weakness 1 & Question 1: Concerns Regarding the Motivation of LRF-SSA
>
> Thank you for your attention to spiking attention mechanisms. We agree with the reviewer’s key observation: the binary spiking nature of SSA indeed leads to information loss, which contributes to the notable performance gap between SNN Transformers and their ANN counterparts. To mitigate this issue, Xiao et al. [1] propose the $\alpha$-XNOR method, which distinguishes the similarity between 0/0 and 0/1 patterns in SSA to improve feature correlation estimation. However, compared with VSA, this method does not account for the inherently **low-entropy nature of attention distributions**, which is crucial for enabling the model to focus on salient regions.
>
> Motivated by this, we introduce a local receptive field mechanism that explicitly leverages the low-entropy property, allowing the model to more robustly attend to relevant regions in the spiking domain and achieve improved attention allocation. To further demonstrate the effectiveness of our method, we conduct experiments on the SpikingFormer architecture [2]:
>
> |Method|Para.|Acc|
> |:-:|:-:|:-:|
> |Spikingformer[2] |29.68|74.79|
> |+$\alpha$-XNOR [1]|29.68|75.39|
> |+LRF-SSA (Ours) |29.71|76.10|
> |+LRF-SSA + $\alpha$-XNOR|29.71|**76.43**|
>
> We observe that both $\alpha$-XNOR and our LRF-SSA module yield clear performance gains, improving accuracy by 0.81% and 1.31%, respectively. Notably, combining the two further boosts performance to 76.43%, representing a 1.64% improvement over the baseline. These results provide additional evidence supporting the effectiveness and complementary nature of our method.
>
> [1] Rethinking Spiking Self-Attention Mechanism: Implementing a-XNOR Similarity Calculation in Spiking Transformers, CVPR 2025.
>
> [2] Spikingformer: A Key Foundation Model for Spiking Neural Networks, AAAI 2026.
>
>
> # Weakness 2: Limited Performance Improvement on ImageNet-1k
> This is primarily because QKFormer already demonstrates strong baseline performance on image classification tasks, leaving limited room for further improvement. To further evaluate its effectiveness, we extend our study to the more challenging image segmentation task, which requires stronger visual long-sequence modeling [3]. The experimental setup follows the Spike-Driven-v3 configuration [4], and the results are presented below:
>
> |Method|Para.|Acc|
> |:-:|:-:|:-:|
> |QKFormer|17.83|34.65|
> |QKFormer+LRF-SSA|17.85|37.19|
> |QKFormer+LRF-Dyn|17.85|37.16|
>
> The experimental results show that both LRF-SSA and LRF-Dyn achieve substantial performance improvements of **2.54%** and **2.51%**, respectively, further validating the effectiveness of the proposed method.
>
> [3] Vision Transformers for Dense Prediction, ICCV 2021.
>
> [4] Scaling Spike-Driven Transformer With Efficient Spike Firing Approximation Training, T-PAMI 2025.

---

> > ### Comment · Reviewer_Ttxv · 2025-11-22
> >
> > Thank you for your response. The concerns I raised in my initial review have been addressed. I believe the authors' response has clarified the paper's motivation. Further experiments have also validated the effectiveness of the proposed method. Therefore, I am willing to raise my rating.

---

### Official Review · Reviewer_Cfzh · 2025-10-29

**Soundness:** 3
**Presentation:** 3
**Contribution:** 3
**Rating:** 6
**Confidence:** 5

**Summary:**

This paper addresses two critical challenges in existing SNN-Transformer architectures: the degradation of local receptive fields and high inference memory cost. The authors present empirical evidence through visualization and statistical analysis, and further propose LRF-Attn and LRF-Dyn to effectively address these issues. Experimental results on both image classification and segmentation tasks demonstrate consistent performance gains, indicating the method’s effectiveness and generality.

**Strengths:**

1. The paper identifies key limitations in existing SNN-Transformer architectures and substantiates these claims through comprehensive empirical observations and quantitative analyses, making the motivation convincing.
2. Extensive experiments are conducted on recent state-of-the-art Transformer architectures, showing consistent and notable improvements in both image classification and segmentation tasks.
3. The authors provide an analysis of the local receptive field degradation in SNNs from both receptive field radius and information entropy perspectives, and convincingly demonstrate the effectiveness of the proposed approach.

**Weaknesses:**

1. The adoption of multi-dendritic neurons increases training complexity, potentially leading to higher computational costs on large-scale datasets such as ImageNet.
2. The presentation of inference memory requirements could be improved. Providing a tabular summary for LRF-Attn and LRF-Dyn would make the comparison clearer and more intuitive.

**Questions:**

1. While the authors have validated the proposed module on mainstream image classification and segmentation tasks, it would be valuable to examine whether comparable effectiveness can be achieved on more challenging NLP tasks [1].
2. Does the proposed LRF-Dyn introduce additional training overhead compared to mainstream CNN-based SNNs [2] and Transformer-based SNNs [3]?
3. Is the local receptive field essential to the effectiveness of the LRF-Dyn method? The authors are encouraged to include experiments to verify this.
[1] Spikelm: Towards general spike-driven language modeling via elastic bi-spiking mechanisms, ICML 2024.
[2] Deep residual learning in spiking neural networks, NeurIPS 2021.
[3] Scaling spike-driven transformer with efficient spike firing approximation training, T-PAMI 2025.

---

> ### Author Response · Authors · 2025-11-22
>
> We sincerely appreciate your recognition of our work. In response to the raised concerns, we provide the following clarifications. All suggested revisions will be incorporated into the revised manuscript to further improve its clarity and presentation.
>
> # Weakness 1 & Question 2: Training Overhead of LRF-Dyn on Large-Scale Datasets and Different Models
> We thank the reviewer for the insightful comment. By leveraging the Fourier transform, we reduce the training complexity of multi-dendritic neurons from $O(N^2)$ to $O(N\log N)$. Specifically, as shown in Eq.15, $\mathrm{sattn}'\_{n}[t]$ during training can be expressed as:
> $$sattn'[t] = [\Gamma, \Gamma \mathcal{A}, ..., \Gamma \mathcal{A}^{k-1}, ..., \tau \mathcal{A}^{n-1}] * [x_1[t], x_2[t], ... , x_{k}[t], ..., x_n[t]] = \mathcal{F}^{-1}(\mathcal{F}(K) * \mathcal{X[t]}), $$
> where $F(\cdot)$ denotes the Fourier transform. During backpropagation, this component incurs a computational complexity of $O(N\log N)$, which is comparable to the $O(N)$ complexity of standard Transformer operations.
>
> To further clarify the training cost, we report the average per-epoch training time on ImageNet under comparable parameter sizes and the same batch size.
> |Method|Para.|Speed|
> |:-:|:-:|:-:|
> |Spikformer [1]|29.68|131.87 samples/s|
> |SEW-ResNet [2]|21|143.68 samples/s|
> |Ours|29.71|126.42 samples/s|
>
> Specifically, Spikformer processes approximately 131.87 samples/s, whereas our method processes approximately 126.42 sample/s.
>
> [1] Spikformer: When Spiking Neural Network Meets Transformer, ICLR 2022.
>
> [2] Deep Residual Learning in Spiking Neural Networks, NeurIPS 2021.
>
> # Weakness 2: Tabulated Comparison of Inference Memory Requirements
> We thank the reviewer for the helpful suggestion, and we will present the results in tabular form. Specifically, we report memory consumption during inference with a batchsize of 16.
>
> | Architecture |          | Spikformer-8-512 |          |          | Spikformer-8-768 |          |          | QKFormer-10-384 |          |          | QKFormer-10-512 |          |          | SDT-V3-S |          |          | SDT-V3-L |          |
> |:-:|:-:|:-:|:-:|:-:|:-:|:-:|:-:|:-:|:-:|:-:|:-:|:-:|:-:|:-:|:-:|:-:|:-:|:-:|
> |    Method    | Baseline |     +LRF-SSA     | +LRF-Dyn | Baseline |     +LRF-SSA     | +LRF-Dyn | Baseline |     +LRF-SSA    | +LRF-Dyn | Baseline |     +LRF-SSA    | +LRF-Dyn | Baseline | +LRF-SSA | +LRF-Dyn | Baseline | +LRF-SSA | +LRF-Dyn |
> |  Memory (MB) |   8.47   |       8.49       |    4.3   |   13.86  |       13.87      |    6.2   |   4.72   |       4.73      |   3.89   |   6.48   |       6.49      |   4.22   |    2.7   |    2.8   |   1.82   |   4.16   |   4.18   |    3.1   |
> |   Acc. (%)   |   73.38  |       74.62      |   74.51  |   74.81  |       75.66      |   75.58  |   78.8   |      79.24      |   79.21  |   82.04  |      82.52      |   82.48  |   75.3   |   76.22  |   76.12  |   79.8   |   80.31  |   80.24  |
>
>
> # Question 1: Assessment of the Modules Effectiveness on Advanced NLP Tasks
>
> To address your concern, we conduct additional experiments on the GLUE dataset, and the results are shown below.
>
> |Method|$\text{MNLI}_{f1}$|QQP|QNLI|SST-2|CoLA|STS-B|MRPC|RTE|Avg|
> |:-:|:-:|:-:|:-:|:-:|:-:|:-:|:-:|:-:|:-:|
> |SpikeLM [3]|77.2|83.9|85.3|87.0|38.8|84.9|85.7|69.0|76.5|
> |SpikeLM-wo/softmax|71.5|58.8|78.6|83.2|23.1|79.2|83.0|65.2|67.83|
> |Ours|73.9|63.5|81.4|85.3|28.2|82.8|84.2|66.4|70.71|
>
> The model with the local receptive field design achieves superior performance, further supporting the effectiveness of our proposed method.
>
> [3] Spikelm: Towards general spike-driven language modeling via elastic bi-spiking mechanisms, ICML 2024.
>
> # Question 2: Evaluating the Role of Local Receptive Fields in LRF-Dyn's Effectiveness
> We emphasize that we conduct a comprehensive ablation study on both the LRF module and the neighborhood size. The corresponding results are presented below.
>
> |Method|w/o LRF| $\Omega \leq 1$|$\Omega \leq 3$| $\Omega \leq 5$|
> |:-:|:-:|:-:|:-:|:-:|
> |LRF-SSA| 77.86 | 78.26 | 78.52 | 78.64 |
> |LRF-Dyn| 77.78 | 78.16 | 78.50 | 78.57 |
>
> We observe that model performance consistently improves as the receptive field expands. To achieve an effective balance between efficiency and performance, we adopt the configuration $\Omega \leq 3$ as our final design choice.

---

### Official Review · Reviewer_LacD · 2025-10-30

**Soundness:** 3
**Presentation:** 3
**Contribution:** 3
**Rating:** 6
**Confidence:** 5

**Summary:**

This paper tackles two critical limitations in Spiking Transformers: the performance gap compared to their Artificial Neural Network (ANN) counterparts and excessive memory overhead. The authors argue these issues stem from the unfocused global attention of Spiking Self Attention (SSA) and the high cost of storing attention matrices. Inspired by biological visual neurons, they introduce LRF-Dyn, a novel method that enhances SSA by integrating a Localized Receptive Field (LRF) mechanism. This addition enables the model to focus on neighboring regions, thereby improving its local modeling capabilities. Furthermore, LRF-Dyn reformulates the attention computation to approximate the charge–fire–reset dynamics of spiking neurons, which significantly reduces memory requirements during inference. Extensive experiments confirm that LRF-Dyn successfully lowers memory overhead while delivering substantial performance gains on visual tasks, establishing it as a key advancement for creating more practical and energy-efficient Spiking Transformers.

**Strengths:**

This paper demonstrates significant strengths across all key dimensions. Its originality is high, stemming from a novel problem analysis that pinpoints the lack of local modeling and high memory cost in Spiking Self Attention (SSA) as key barriers . The proposed LRF-Dyn method is a creative and well-motivated solution, uniquely integrating a Localized Receptive Field (LRF) mechanism to enhance local feature capture and, more innovatively, reformulating the attention mechanism to mimic neuronal dynamics, thereby reducing memory overhead. The quality of the work is excellent, substantiated by a solid theoretical foundation, including theorems that formally prove LRF-SSA preserves the desirable local attention and low-entropy properties of standard attention mechanisms . These theoretical claims are rigorously supported by extensive experiments that confirm both performance improvements and reduced memory consumption. The paper is presented with outstanding clarity, logically progressing from a well-defined problem to a tailored solution, with concepts effectively illustrated through figures and formal proofs. Finally, the significance of this work is substantial; by directly tackling the critical performance and efficiency bottlenecks of Spiking Transformers, it provides a practical and effective component that could accelerate the deployment of energy-efficient vision models on resource-constrained edge and neuromorphic hardware.

**Weaknesses:**

1. To substantiate the claimed benefits of the proposed method, such as the reduction in memory overhead, the authors should provide concrete quantitative data, for instance, measurements of energy consumption and inference latency.
2. To further justify the necessity of multi-dendritic neurons, the authors are encouraged to include comparisons under identical conditions with other spiking neuron (e.g., LIF, ALIF or DH-LIF) that serve as the core neurons in LRF-Dyn.
3. Given that LRF-Dyn exhibits more complex dynamical behavior, the authors should consider providing an algorithmic description or pseudocode to facilitate a clearer understanding of the proposed method.

**Questions:**

1. Under identical configurations, how does LRF-Dyn perform when employing different spiking neuron such as LIF, ALIF or DH-LIF?
2. Can the proposed module maintain its effectiveness and generality when evaluated on additional neuromorphic datasets, such as CIFAR10DVS?
3. How does the number of dendrites affect the model's performance and results?

---

> ### Author Response · Authors · 2025-11-21
>
> We sincerely appreciate your recognition of our work. In response to the questions, we provide the following clarifications. All suggested revisions will be incorporated into the revised version to further enhance its clarity and presentation.
> # Weakness 1: Quantitative Validation of Energy Consumption, Memory Usage, and Inference Latency Benefits
>
> We appreciate the reviewer’s insightful comment. We would like to clarify that our original manuscript already reports the inference-time memory requirements of the compared models. As shown in Fig. 5(b), the bubble size represents the peak memory usage during inference, while the horizontal and vertical axes denote the model parameter scale and performance, respectively.
>
> To more clearly demonstrate the effectiveness of the proposed method, we further conduct quantitative evaluations of **energy consumption**, **memory usage**, and **inference latency** under a consistent batch size of 16 across multiple model settings.
> | Architecture |          | Spikformer-8-512 |          |          | Spikformer-8-768 |          |          | QKFormer-10-384 |          |          | QKFormer-10-512 |          |          | SDT-V3-S |          |          | SDT-V3-L |          |
> |:-:|:-:|:-:|:-:|:-:|:-:|:-:|:-:|:-:|:-:|:-:|:-:|:-:|:-:|:-:|:-:|:-:|:-:|:-:|
> |    Method    | Baseline |     +LRF-SSA     | +LRF-Dyn | Baseline |     +LRF-SSA     | +LRF-Dyn | Baseline |     +LRF-SSA    | +LRF-Dyn | Baseline |     +LRF-SSA    | +LRF-Dyn | Baseline | +LRF-SSA | +LRF-Dyn | Baseline | +LRF-SSA | +LRF-Dyn |
> Energy (mJ) | 7.73 | 7.74| 6.93| 21.48| 21.51| 20.12| 15.13| 15.14| 14.23| 21.99| 22.01| 20.56| 1.70| 1.71| 1.56| 5.90| 5.91| 5.05|
> | Latency (ms) |   38.5   |       39.6       |   42.2   |   72.1   |       72.2       |   72.6   |   27.8   |       28.1      |   28.3   |   33.8   |       34.2      |   34.3   |   13.6   |   13.8   |   13.9   |   22.1   |   23.2   |   23.4   |
> |  Memory (MB) |   8.47   |       8.49       |    4.3   |   13.86  |       13.87      |    6.2   |   4.72   |       4.73      |   3.89   |   6.48   |       6.49      |   4.22   |    2.7   |    2.8   |   1.82   |   4.16   |   4.18   |    3.1   |
> |   Acc. (%)   |   73.38  |       74.62      |   74.51  |   74.81  |       75.66      |   75.58  |   78.8   |      79.24      |   79.21  |   82.04  |      82.52      |   82.48  |   75.3   |   76.22  |   76.12  |   79.8   |   80.31  |   80.24  |
>
> Here, **Energy** denotes the theoretical energy consumption during inference. **Memory** denotes the peak inference-time memory usage, and **Latency** refers to the average time required to process a single image. As shown in the results, RF-SSA enhances model performance, whereas LRF-Dyn markedly reduces memory consumption without incurring accuracy loss.
>
>
> # Weakness 2 & Question 1: Performance of LRF-Dyn with Different Spiking Neuron Models
> To address your concern, we only replace the multi-dendritic neurons in LRF-Dyn and conduct experiments on the SDT-V3 dataset. The results are as follows:
> |Method|Acc|
> |:-:|:-:|
> |LIF-based|65.80|
> |ALIF-based [1]|70.25|
> |DH-LIF-based [2]|72.51|
> |**LR-Dyn (Ours)**|**76.12**|
>
> As shown in the table, under the same architectural design, point neurons LIF and ALIF achieve only 65.80% and 70.25% accuracy, respectively. In contrast, DH-LIF and LR-Dyn reach 72.51% and **76.12%**, demonstrating notably superior performance. This further highlights the necessity of multi-dendritic neuron modeling.
>
> [1] Accurate and efficient time-domain classification with adaptive spiking recurrent neural networks, Nature Machine Intelligence, 2021.
>
> [2] Temporal dendritic heterogeneity incorporated with spiking neural networks for learning multi-timescale dynamics, Nature Communications, 2024.
>
> # Question 2: Evaluating Generality on Additional Neuromorphic Datasets?
> To further validate the effectiveness of the proposed method, we conduct experiments on the CIFAR10-DVS dataset, using the same model configuration as in Spikformer[1].
> |Method|Acc.|
> |:-:|:-:|
> |Spikformer [3]| 80.9 |
> |LRF-Attn (Ours) | 82.4 |
> |LRF-Dyn (Ours) | 82.1 |
>
> [3] Spikformer: When Spiking Neural Network Meets Transformer, ICLR 2022.
>
> # Question 3:  Effect of Dendrite Number on Performance
> We employ the SDT-V3 architecture and evaluate it on the ImageNet dataset. The experimental results are presented below.
>
> |Number of Dendritc|Para.|Acc|
> |:-:|:-:|:-:|
> |n=4|3.83|74.20|
> |n=8|5.21|76.12 |
> |n=16|5.43|76.25|
>
> The results indicate that model performance improves as the number of dendrites increases. To balance energy efficiency and training costs, we set the dendrite number to n = 8 in the final configuration.

---

> ### Comment · Reviewer_LacD · 2025-11-24
> **Response to the Rebuttal**
>
> Thank you for the detailed reply; it has clarified my concerns. Taking the rebuttal into account, I find the motivation sound and the experiments more reliable, so I will raise my score.

---

### Author Response · Authors · 2025-12-03

Dear ACs,

We sincerely appreciate your time and effort in handling our submission. We also thank all reviewers for their constructive feedback, which has substantially improved the quality of our manuscript. Overall, the reviewers expressed clear recognition of our work, particularly in the following aspects:

* **Clear problem identification and methodological innovation:** The manuscript identifies the core limitations of SSA and introduces the original LRF-SSA and LRF-Dyn mechanisms. (Reviewer LacD, Reviewer Cfzh, Reviewer MjDS)

* **Solid theoretical foundations:** The proposed methods are supported by solid theoretical analysis, with well-defined connections to conventional attention formulations [1]. (Reviewer LacD, Reviewer Cfzh, Reviewer Ttxv, Reviewer MjDS)

* **Comprehensive and diverse empirical validation:** Extensive experiments across multiple architectures and tasks demonstrate the effectiveness and practical value of the proposed approach. (Reviewer LacD, Reviewer Cfzh, Reviewer MjDS)

In response to the reviewers’ comments, we have addressed the main concerns as follows:

* **Quantitative analysis of training and inference costs:** We quantify training complexity, memory usage, inference latency, and training throughput under matched parameter budgets, further confirming the method’s efficiency.

* **Concerns regarding limited performance gains and method motivation:** We clarify the theoretical motivation by analyzing the low-entropy attention distribution caused by binary spiking, demonstrating the necessity of localized receptive fields. Additional ablation with $\alpha$-XNOR [2] and more challenging segmentation experiments (showing >2.5% gains) further validate the motivation and effectiveness of the proposed modules. **These explanations are acknowledged by Reviewer Ttxv**.

* **Verification of scalability and practical applicability:** We expand our experiments to neuromorphic datasets, NLP benchmarks, long-sequence tasks, and segmentation tasks, and include analyses of training stability, computational complexity, and comparisons with mainstream SNN/CNN/Transformer models. The results confirm strong scalability and stability across diverse settings.


Reviewer LacD, Reviewer Cfzh, and Reviewer MjDS **express positive assessments** in the initial round. Notably, Reviewer Ttxv upgrades the rating **from 2 to 6**, acknowledging that our rebuttal effectively addressed their initial concerns. Reviewer LacD further increases their score **from 6 to 8** after our detailed responses.

We sincerely appreciate the reviewers’ thoughtful feedback and have incorporated all suggestions into the revised manuscript. Thank you again for your time and consideration.

[1] Spikformer: When Spiking Neural Network Meets Transformer, ICLR 2022.

[2] Rethinking Spiking Self-Attention Mechanism: Implementing a-XNOR Similarity Calculation in Spiking Transformers, CVPR 2025.

---

### Meta-Review · Area_Chair_QPGR · 2026-01-02

**Summary:**

I am not sure what to put here as an answer about "Provide a summary of the reviewers' concerns that informed your suggested decision for this paper.", when the conerns were addressed and it is now a clear accept. In fact, the concerns were not of a major type and more questions, which then the authors did satisfactorily answered such as "The motivation behind LRF-SSA requires further validation." (Reviwer Ttxv) and of a similar type.

**Reviewer Concerns:**

To my mind, the reviewers concerns were satisfactorily addressed.

**Reviewer Scores:**

It’s really hard to say how any reviewer would have changed their score if they had taken part more fully in the discussion. Without hearing it from them directly, anything we write here would just be guesswork.

For this paper, the scores were 6,6,2,6. Reviewer Ttxv with the grade 2 replied "Thank you for your response. The concerns I raised in my initial review have been addressed. I believe the authors' response has clarified the paper's motivation. Further experiments have also validated the effectiveness of the proposed method. Therefore, I am willing to raise my rating.".

Hence it is now a clear accept.

---

### Decision · Program_Chairs · 2026-01-26

Accept (Poster)